# First Integrative Morphological and Genetic Characterization of *Tremoctopus violaceus*
*sensu stricto* in the Mediterranean Sea

**DOI:** 10.3390/ani12010080

**Published:** 2021-12-30

**Authors:** Blondine Agus, Pierluigi Carbonara, Riccardo Melis, Rita Cannas, Laura Carugati, Jacopo Cera, Marilena Donnaloia, Antonello Mulas, Antonio Pais, Stefano Ruiu, Giuseppe Vinci, Danila Cuccu

**Affiliations:** 1Department of Life and Environmental Science, University of Cagliari, 09126 Cagliari, Italy; blondine.agus@unica.it (B.A.); riccardo.melis@unica.it (R.M.); rcannas@unica.it (R.C.); laura.carugati@unica.it (L.C.); jacopo.cera96@gmail.com (J.C.); amulas@unica.it (A.M.); stefanoruiu96@gmail.com (S.R.); 2Coispa Tecnologia & Ricerca Stazione Sperimentale per lo Studio delle Risorse del Mare, 70126 Bari, Italy; carbonara@coispa.it (P.C.); donnaloia@coispa.eu (M.D.); 3Department of Agriculture, University of Sassari, 07100 Sassari, Italy; pais@uniss.it; 4Parco Regionale delle Dune Costiere, Ostuni, 72017 Brindisi, Italy; giuseppevinci15@gmail.com

**Keywords:** *Tremoctopus violaceus*, molecular analysis, biometric features, age, mating

## Abstract

**Simple Summary:**

Four rare species are recognized within the genus *Tremoctopus* (Cephalopoda: Octopoda), i.e., *T. gelatus*, *T. gracilis*, *T. robsoni*, and *T. violaceus*. The accurate identification of organisms is a fundamental prerequisite to deepen our knowledge of the biology and ecology of a species. In this study, for the first time, an integrative morphological and genetic approach was undertaken to confirm the identity of specimens of the genus *Tremoctopus* collected in Mediterranean waters. Sequences of two mitochondrial genes were generated and analyzed from three Mediterranean females, allowing ascribing all the samples to the species *T. violaceus sensu stricto*. For the first time, barcoding sequences have been obtained from the presumed type locality of the species. This information is of particular importance for this rare species; it has been complemented with the detailed descriptions of morphometric and biological features, as well as beaks analyses for the age estimation of the samples.

**Abstract:**

An integrative approach based on morphological and genetic analyses was undertaken for the first time to confirm the species identification of Mediterranean samples belonging to the genus *Tremoctopus*. Sequences of two mtDNA genes (cytochrome c oxidase subunit (COI) and 16S) were generated for the first time from Mediterranean samples. Both the similarity-based identifications and tree-based methods indicated that three females can be identified as *Tremoctopus violaceus sensu stricto* in agreement with their morphological classifications. All Mediterranean sequences clustered with the sequences of *Tremoctopus violaceus* from the Gulf of Mexico and were clearly differentiated from the sequences attributed to *T. gracilis* and *T. robsoni*. The chromatic pattern of the web and some features of gill filaments, arms formula, stylets, radulae, beaks, and stomach contents were given for all the samples; 105,758, 20,140, and 11,237 oocytes were estimated in the mature, immature, and developing samples, respectively. The presence of four spermatangia inside the cavity of the maturing female suggested the ability of this species to mate before reaching full maturity with more partners. Age investigation using beaks, performed for the first time in *T. violaceus* and within the genus gave results consistent with the different sizes and maturity conditions of the samples.

## 1. Introduction

Species of the genus *Tremoctopus* Delle Chiaje, 1830 (Cephalopoda: Octopoda), commonly known as “blanket octopods” on account of the females-expanded dorsal web that unites the dorsal arms, where the eggs are brooded until they hatch [1,2], are rarely encountered, since they spend their entire life cycle in the open ocean. They are characterized by marked sexual size dimorphism, with small dwarf males and large females, some of which reach 2 m in length [3,4]. Four species are recognized as valid within the genus *Tremoctopus*, i.e., *Tremoctopus gelatus* Thomas, 1977, which is meso-bathypelagic with circumtropical and temperate distributions, *Tremoctopus gracilis* Souleyet, 1852 that occurs in the Pacific and Indian oceans, *Tremoctopus robsoni* Kirk, 1884 described from waters off New Zealand, and *Tremoctopus violaceus* originally portrayed in the Mediterranean waters by Delle Chiaje (1830) and distributed also in the Atlantic Ocean including the Gulf of Mexico and the Caribbean Sea [5,6,7]. Apart from *T. gelatus*, of which the males and females are easily recognized being characterized by their gelatinous tissues, separating the other three species is not an easy task. The counts of the proximal and distal suckers of the hectocotylus are used for the species identification of males. On the contrary, females are recognized by less precise criteria: *T. robsoni* for their long and convoluted distal oviducts, while adults of *T. violaceus* and *T. gracilis* for the different chromatic patterns of the web surrounding the dorsal arms. However, *T. violaceus* and *T. gracilis* are reported to be almost indistinguishable in their juvenile phase [8,9].

The difficulty in separating these taxa, as well as the paucity of molecular data, could cause several errors in identifications, especially involving *T. violaceus* and *T. gracilis*, as exemplified by the records of these two species outside their known geographical limits [7]. For instance, in the Mediterranean Sea, in the past, *T. violaceus* was the only member of the genus known to live in the area, while nowadays it is reported that two species of the genus *Tremoctopus* occur in the Mediterranean Sea, i.e., *T.*
*violaceus* as the native species and *T. gracilis* as an alien species probably arriving in the Mediterranean due to human-mediated transfer or with Lessepsian migrations [8,10,11].

Most of the Mediterranean records of *Tremoctopus* sp., often classified as *T. violaceus*, refer to individuals stranded [12,13,14,15] or remain found in stomach contents of large pelagic predators such as swordfish *Xiphias gladius* Linnaeus, 1758 [16]. However, the occurrence of *T. gracilis* was reported in the Mediterranean Sea as early as in 1936 in the northern Adriatic Sea [17]; the individual, originally identified as *T. violaceus*, was later reclassified as *T. gracilis* based on the color pattern [10]. Recently, in August 2002, a large egg-carrying female of *Tremoctopus* sp. was observed and photographed in the waters of Ponza Island in the Tyrrhenian Sea [18] and attributed to *T. gracilis* on the basis of the chromatic pattern of the web [8,9]. This specimen, indeed, exhibited dorsal spots different from the large round spots considered typical of *T. violaceus*. In recent years, *T. gracilis* has been further recorded in Tunisian waters [19,20]. It cannot be excluded that other *T. gracilis* are misidentified with *T. violaceus* in the past, since *T. violaceus* is the only member of the genus known to live in the Mediterranean [10]. All these Mediterranean records of *T. violaceus* and *T. gracilis* refer to female specimens identified only by morphological characters that have been reported to be not always solid enough to avoid misidentifications [7]. Indeed, no genetic analyses have been conducted to date on the genus *Tremoctopus* in the Mediterranean Sea, and no sequences are available yet in the public repositories from this area.

In this paper, we aimed to identify and characterize three new specimens collected in the Mediterranean Sea, belonging to the genus *Tremoctopus*, with an approach based on a combined morphological and molecular analysis, and the latter represented the first genetic analysis on the genus *Tremoctopus* performed in the Mediterranean Basin. Newly generated DNA sequences were used to support the morphological identification and compared with those available from other areas, to better characterize the intra- and interspecific variability within the genus. Furthermore, detailed morphological and meristic features were provided for the studied specimens, and for the first time, an age investigation has been performed to analyze their beaks.

## 2. Materials and Methods

### 2.1. Morphometric and Biological Analysis

Three studied specimens were found in the Mediterranean Sea, with two in Sardinian waters (from henceforth TV11 and TV16) and one in the Adriatic Sea (TVBA). The first was found by a fisherman in an octopus trap at a depth of 6 metres, and the other two were accidentally collected, stranded inshore and given to the research institute by citizens. In the laboratory, the dead specimens were classified [2], measured (to the nearest 0.1 mm), weighed (to the nearest 0.01 g) and sexed. Biometric parameters and indexes were taken and calculated as following [21]. The mantle cavities were inspected to count the gill lamellae and to establish the maturity condition following the scale at five stages (immature, developing, maturing, mature/spawning, and spent) currently in use for cephalopods within the MEDITS project [22,23]. The presence of spermatophores/spermatangia was recorded, as a sign of mating occurred. The ovary oocyte sizes (the major axis was measured to the nearest 0.01 mm) were taken, and the potential fecundity was gravimetrically estimated as the sum of the total oocytes number (>0.05 mm) [24]. The stomachs were removed, and the contents were assigned to a subjective fullness index (FUI), where 0 indicates the content is empty, 1 indicates the content containing very scarce remaining, and 2 indicates the content from significant remains to full repletion [25]. Stomach contents were used for prey identification at a category level. Entire stylets and beaks were extracted and measured in all the samples. According to [26], standard measurements of the upper and lower beaks were taken using a digital calliper (accurate to one tenth of a millimetre). With the aim to estimate the age, according to [27], upper beaks were sectioned sagittally in two symmetrical pieces along the posterior edge of the hood and crest, to count increments, assuming a daily deposition. Increments were counted five times by the same operator at different times, considering the average of readings. To avoid duplicating counts, each increment was numbered using the Tps_Dig2 software (Figure 1J). The overall readings precision and accuracy were evaluated by the coefficient of variation (%CV) [28] and the average percent error (APE) [29] calculated as:$$APE_{J}=100\%\;\times \;\frac{1}{R}\;\sum_{I=1}^{R}\frac{\left|X_{ij}-X_{j}\right|}{X_{J}}c,$$
where *X_ij_* is the *ith* age determination of the *jth* fish, *X_j_* is the mean age estimate of the *jth* fish, and *R* is the number of times each fish is aged.

### 2.2. DNA Extraction, Amplification, and Sequencing

Total genomic DNA was extracted from the arm or mantle tissues of the three specimens using a PureLink™ Genomic DNA kit (Invitrogen, Thermo Fisher Scientific Inc., Dublin, UK), following the manufacturer’s protocol. The extracted DNA amount and the quality determination were evaluated using the agarose gel electrophoresis and a NanoDrop™ One (Thermo scientific, Thermo Fisher Scientific Inc., Dublin, UK) UV spectrophotometer. The primers for the amplification of the mitochondrial cytochrome c oxidase subunit I (COI) and mitochondrial 16S rRNA (16S) genes were obtained from [30,31], respectively. The amplification of the COI gene was based on the following cycling parameters: initial denaturation at 95 °C for 2 min, followed by 43 cycles of denaturation at 95 °C for 30 s, annealing at 45 °C for 30 s, and extension at 72 °C for 45 s, with a final extension of 5 min at 72 °C. Cycling conditions of 16S only differed for the number of cycles (38) and the annealing temperature (49 °C). The PCRs were set up in a 25 μL reaction volume containing 2.5 μL of 10X buffer (Dream Taq^®^ buffer, Thermo scientific, Thermo Fisher Scientific Inc., Dublin, UK), 2.5 μL of 2 mM dNTPs, 1.5 μL of 25 mM of MgCl_2_, 0.2 μL of each 20 mM primer, 0.16 μL of Taq polymerase (Dream Taq^®^ Thermo scientific, Thermo Fisher Scientific Inc., Dublin, UK), and 2 μL of DNA (50–100 ng). The PCR products were sequenced by Macrogen Europe (Amsterdam, The Netherlands).

### 2.3. Molecular Analyses

The Clustal W algorithm [32] implemented in MEGA v7 [33] was used to align the sequences. The genetic identification of the three individuals was based on two different approaches, i.e., a similarity-based method and a tree-based method. The first procedure was based on the BLASTn search routine implemented in GenBank using the default parameters (https://blast.ncbi.nlm.nih.gov, accessed on 30 August 2021). This method compared our sequences with those available in the database and provided a list of similar sequences and the respective percentual similarity. Analogously, our COI sequences were compared to all available barcode records in BOLD (https://www.boldsystems.org, accessed on 30 August 2021), using the identification engine BOLD-IDS, with the option “COI Full Database”, which included also records without species designation. No 16S database is available for BOLD-IDS. In order to assess the relationship between our query sequences and its neighboring reference sequences, all the COI and 16S sequences were separately analyzed using tree-based approaches: the Bayesian inference (BI), using the software MrBayes v3.2.7 [34], and the maximum likelihood (ML) and neighbor-joining (NJ) methods, implemented in MEGA. Each COI and 16S sequence produced in this study were included in a phylogenetic analysis with homologous sequences, belonging to the genus *Tremoctopus*, available in the public repositories (GenBank and BOLD; Appendix A). Sequences derived from a complete mitochondrial genome (AB158363; [35]), attributed to the species *Octopus vulgaris*, were used as an outgroup for both COI and 16S markers. The distances among the *Tremoctopus* groups identified in the trees were calculated in MEGA in the Kimura two-parameter model [36], which is the most widely used in barcoding studies, following the suggestion by [37].

## 3. Results

### 3.1. Morphometric and Biological Results

The specimens exhibited a dark bluish-purple body, a head and iridescent silvery ventral surface, four water pores (two dorsal and two ventral pores) and a web surrounding the dorsal arms (even if broken) (Figure 1A–C). The arms were unequal in length and shape; despite most of them being truncated or heavily damaged, they seemed to follow the formula 2,1,4,3. All samples had 13 gill lamellae, transparent stylets and beaks characterized by the absence of the rostrum, with a pigmentation limited in the crest and hood (Figure 1G,H; Table 1). Their radula had seven teeth as well as two thin, rectangular marginal plates per transverse row with a tricuspid rachidian tooth (Figure 1I,K). All these features led to classifying three females of *Tremoctopus violaceus* according to [2]. As regards the colour pattern of the web, the samples showed a series of patches forming clusters arranged transversely (Figure 1A), with most visible in fresh animals. This web pattern was not correspondent with the description proposed for *T. violaceus* [8,9].

Apart from the biggest female (TV11 with 165 mm in the mantle length and 875 g in the total weight) that was close to being sexually mature but with empty oviducts (Figure 1D), the other two samples (TV16 and TVBA) were at the preliminary phases of the sexual maturation process. All ovaries were composed of reticulated oocytes variable in size with an average of 0.3 mm (Figure 1E). The values of fecundity estimated were 105,758, 20,140, and 11,237 oocytes in the three samples, respectively. (Table 1). The maturing female already mated because of the presence of four spermatangia in its ventral cavity (Figure 1D,F). The stomachs of the specimens TV16 and TVBA had remains of *Posidonia oceanica* and those of species belonging to the phylum Entoprocta. The upper beaks showed a distinct band pattern (increments) (Figure 1J), until a maximum number of 161 days was counted in the biggest female. The low values of APE and CV obtained for all analyses showed high levels of precision in the readings (Table 1).

### 3.2. Molecular Analysis

The newly obtained sequences of 16S and COI were 517 and 616 bp long, respectively (GenBank accession numbers: OM025092-4 and OM025233). However, the low quality and quantity of the extracted DNA, due to the poor condition of the tissue samples, resulted in difficulties in the amplification and sequencing of the COI gene. In particular, COI amplification failed in TV11, while it produced a highly divergent “COI-like” sequence being 604 bp long in TV16, of which the translation into the aminoacidic sequence revealed stop codons and indels suggesting that could be a pseudogene. Therefore, the COI obtained for TV16 was excluded from the subsequent analyses.

To assess the relationship between our query sequences and their neighboring reference sequences, each COI and 16S sequence produced in this study were included in a phylogenetic analysis with the homologous sequences, belonging to the *Tremoctopus* genus, available in the public repositories. Considering all the *Tremoctopus* sequences, the final alignments comprised 10 sequences for COI and 7 sequences for 16S, including the sequences derived from a complete mitochondrial genome (GenBank Accession KY649286) used for both the markers (Appendix A).

As regards the 16S gene, our three individuals shared the same haplotype; the BLASTn comparison returned a match with an individual of *T. violaceus* from the Atlantic waters of Mexico (GenBank accession number: MT271737), with a similarity of 99.8%. The nearest compatibility (93.5–93%) resulted with a group of three sequences (MN435565, KY649286, and AJ252767), with all deposited under the name *T. violaceus*. The 16S trees, based on a low number of public sequences, showed two supported groups, i.e., clades A and B (Figure 2). Our three sequences clustered together in clade B with an individual of *T. violaceus* from the Gulf of Mexico were well separated from the other three sequences available in GenBank (accession numbers: MN435565, KY649286, and AJ252767), grouped in clade A. As already pointed out by [7], the sequences in clade A were likely misidentified and corresponded to *T. gracilis*. The estimate of the divergence between the groups resulted in a 6.4% distance, while very slight differences were found among the sequences within the clades (0.6% divergence in clade A and 0% in clade B; Table 2).

Using the BLASTn search implemented in GenBank, the COI sequence had the better match with the sequence having the GenBank accession number of KY649286 attributed to the species *T. violaceus*, with a similarity of 92.05%. The BOLD-IDS showed that TVBA had a very high correspondence (similarity: >99.7%) with three private sequences (BOLD accession numbers: FLBAR686-18, FLBAR760-18, and FLBAR687-18), from Florida (USA) in the Gulf of Mexico, attributed to the species *T. violaceus* (BIN BOLD:ADM8329); unfortunately, they are not available in the public database, so they cannot be included in the phylogenetic analysis. In the BI, ML, and NJ trees, COI sequences clustered in three highly supported clades—A, B, and C (Figure 3). The Mediterranean individual obtained in this study separately clustered in clade B, clade A grouped the sequences from Pacific, Atlantic, and Indian Oceans, and clade C comprised two sequences from New Zealand and Chilean individuals. Clade A sequences corresponded to *T. gracilis*, clade B corresponded to *T. violaceus*, and clade C was ascribed to *T. robsoni*. It is worth stressing that, as in the case of 16S, also for COI gene, some of the sequences deposited in public repositories under the name of *T. violaceus* were actually *T. gracilis*. The COI distances among the clades reflected the topography of the trees, with the maximum distance (about 14%) between clades A and B in comparison with clade C, and a 9% distance between the nearest groups (Table 2). The intra-groups distances highlighted the homogeneity within clade A (divergence: 0.2%) and clade C (divergence: 0%).

## 4. Discussion

### 4.1. Morphometric and Biological Results

The scarcity of the captures prevents the full understanding of the life-history of *Tremoctopus violaceus*, especially in the Mediterranean Sea. Therefore, morphological and biometrical information obtained in this study, even if with the limitation of the sample size, could contribute to filling our gaps of knowledge regarding this species. The absence of an entire web in all the females analyzed confirmed the significance of the autotomy as a defensive tool by means of the detachment of segments [38]. The chromatic pattern of the web has been described with large round spots in *T. violaceus* [1,7,9] and with series of patches that form clusters arranged transversely with respect to the longitudinal axis of the arm in *T. gracilis* [9,17]. In our samples, the chromatic pattern, even if observed only in the remains of pieces of webs, seemed to match the second case and to be similar to the webs figured by [17,39] and photographed in the coastal water of Ponza Island (in [9]). This finding opens the question if the web chromatic pattern is a valid diagnostic character to discriminate the two species *T. violaceus* and *T. gracilis*. In this study, indeed, the colour pattern of *T. gracilis* seemed to be present in specimens that are molecularly identified as *T. violaceus*.

Most of the findings reported until now in both the western and eastern parts of the Mediterranean, regarded female adults accidentally caught or stranded/moribund in coastal waters, mostly in summer [12,13,14,15]. According to [40], this species is known to be epipelagic, approaching coastal waters for reproduction showing an increased vulnerability to fishing or stranding. Considering that *T. violaceus* is reported to feed on pteropod molluscs and small fishes [5], the presence of this species in shallow water could explain also the stomach contents such as *Posidonia oceanica* and sessile benthic organisms as observed in our two samples and previously reported also for the mature female stranded on a beach of the Balearic Island [40]. In the literature, a variable number of gill filaments (13–16) have been reported for females of this species and at least two different arms orders (2.1.4.3; 1.2.4.3). In our observations, we always counted 13 gill filaments, and even if the arms were damaged to some degree, they seemed to follow the formula 2.1.4.3 as the female from the Gulf of Mexico was recently observed, confirmed being *T. violaceus sensu stricto* [7]. The body proportions of the samples, as well the features of their beaks (upper and lower), stylets, and radulae, are in agreement with the literature [2,5,7]. Laptikhovsky and Salman [14] classified the species as an intermittent terminal spawner based on the analysis of five maturing, mature, and spawning females from the Aegean Sea. The size of the ovary oocytes of our biggest female resulted smaller when compared with the Aegean ones at the same maturity stage, while the value of potential fecundity was close to the lower limit of the referential range [14,15]. Despite the presence of still small and reticulated oocytes in the ovary as well as the total absence of ripe ones, the observation of spermatangia inside the mantle cavity of the maturing female shows the ability of the species to mate before reaching full maturity, like most coleoid cephalopods. Thomas [2] stated: “Tremoctopus male autotomizes its hectocotylized arm during mating [41]. Sexually mature females are frequently found with these detached arms (often more arm per female) lying in the mantle cavity”. Honestly, by our single observation we cannot give more details on spermatangia, apart from their morphology and that they were not inside oviducal glands/oviducts, as typically known for most octopods (e.g., [42,43,44]). Moreover, because it is known that males can produce typically one spermatophore [2], the presence of four spermatangia in our female suggests a multiple mating. As regards the age, assuming daily increments in the upper beak, as observed and validated in other octopods [27,45], we here reported the first attempt of age estimation using beaks, and our results are consistent to their different sizes and maturity conditions of the investigated specimens.

### 4.2. Molecular Analysis

For the first time in this work, specimens of the genus *Tremoctopus* caught in the Mediterranean waters were genetically characterized using two mitochondrial fragments, the COI and the large ribosomal subunit 16S. Both markers are widely used and proved to be useful tools for identifying cephalopod species [7,37,46,47,48,49,50].

Both the similarity-based identification and the tree-based method showed that the specimens caught in Mediterranean waters belonged to the same group, ascribable to the same species, i.e., *T. violaceus sensu stricto*. Although reference [51] did not formally designate a type locality of *T. violaceus*, the original description was based on specimens caught into the Kingdom of Naples, and consequently, the Mediterranean waters can be considered the type locality of the species. In the end, our work represents the first genetic investigation and the first available sequences for the type locality of *T. violaceus*.

The comparison of the Mediterranean and homologous 16S sequences mined from public databases showed close similarity between our investigated specimens and individuals found in the southwestern Gulf of Mexico, confirming the attribution to the species *T. violaceus sensu stricto* [7]. 16S sequences from the Pacific individuals showed a level of divergence from the Atlantic and Mediterranean samples compatible with two distinct species and were attributed to *T. gracilis* [7], despite being deposited under a different name. Similarly, the COI tree showed the specimen from the Mediterranean (*T. violaceus sensu stricto*) clearly separated from the Pacific, Atlantic, and Indian Oceans individuals (*T. gracilis*, deposited as *T. violaceus*) and two individuals coming from New Zealand and Chile that should be ascribed to the species *T. robsoni*, as identified by [52]. For both genetic markers, COI and 16S, sequences deposited under the name *T. violaceus* proved to be *T. gracilis*. This study once again confirms that errors/misidentifications are common in public repositories [7,53], probably because there are not many and easily applicable taxonomic characters used in the field to distinguish them, especially separating *T. violaceus* and *T. gracilis*. These discrepancies between the morphological and molecular identification, suggest adopting a more precautionary approach when reassigning old records to a different species based solely on the colour patterns deducted from drawings/photos [8,9,10] or extending the geographical range of species (e.g., *T. gracilis* within the Mediterranean Sea), based solely on sporadic observations, described in conference papers almost impossible to find for consultation [19].

Difficulties encountered for COI amplification in the Sardinian specimens were not unexpected considering the poor condition of the tissues and suggested that for rare species, special attention should be given to the immediate preservation of the tissue used for genetic analyses. Similarly, the finding of COI-like sequences was not surprising, since they are increasingly reported in major clades characterized by large genomes, which contain a high frequency of nuclear pseudogenes originating from the mitochondrial genome (nuclear DNA of mitochondrial origin, nuMTs). In some crustaceans, the COI gene, typically used in DNA barcoding, was found in disproportionally higher diversity and coverage in the nuclear DNA than the rest of the mitogenome, consistent with multiple insertions of that region into the nuclear genome [54]. In the future, different markers (less likely to be duplicated as nuMTs) or species-specific primers should be used to further characterize *Tremoctopus* specimens from Sardinia, in order to additionally check for the genetic variability of *T. violaceus* within the Mediterranean Basin.

## 5. Conclusions

To date, our study represents the first contribution to the genetic characterization of the species of genus *Tremoctopus* in the Mediterranean Sea. We obtained the first sequences of two mtDNA genes (COI and 16S), ascribed to the species *T. violaceus*, from individuals caught in the type location of the species. Further analyses, with a higher number of specimens and additional markers, are needed to investigate the genetic differentiation in the Mediterranean Sea. Besides, this work highlights the scarcity of genetic data and the occurrence of erroneous records in international repositories, particularly in this genus where most of the few sequences publicly available are reported under the identical name, *T. violaceus*. Once more, we reaffirmed the urgent need to check and keep updating the information reported in the public databases, in order to reduce the confusion in the taxonomy of the species. Considering this background and the findings obtained in this study by an integrative approach, we confirmed that the routine integration of genetics into future investigations within the genus is highly recommendable. Its use could be fundamental to confirm the occurrence of *T. gracilis* in the Mediterranean and its establishment success in these waters and to fill in the gaps in the knowledge of the genus in this area. However, the scarcity of the captures of *Tremoctopus* species still represents the main difficulty to understand fully the life-history of the species. In this context, the collaboration of citizens, as for our study, could be very useful to improve the collection of specimens and data. Moreover, scientists should benefit from the limited and fragmentary information reporting sporadic findings, as performed in the present study for *T. violaceus*.

## Figures and Tables

**Figure 1 animals-12-00080-f001:**
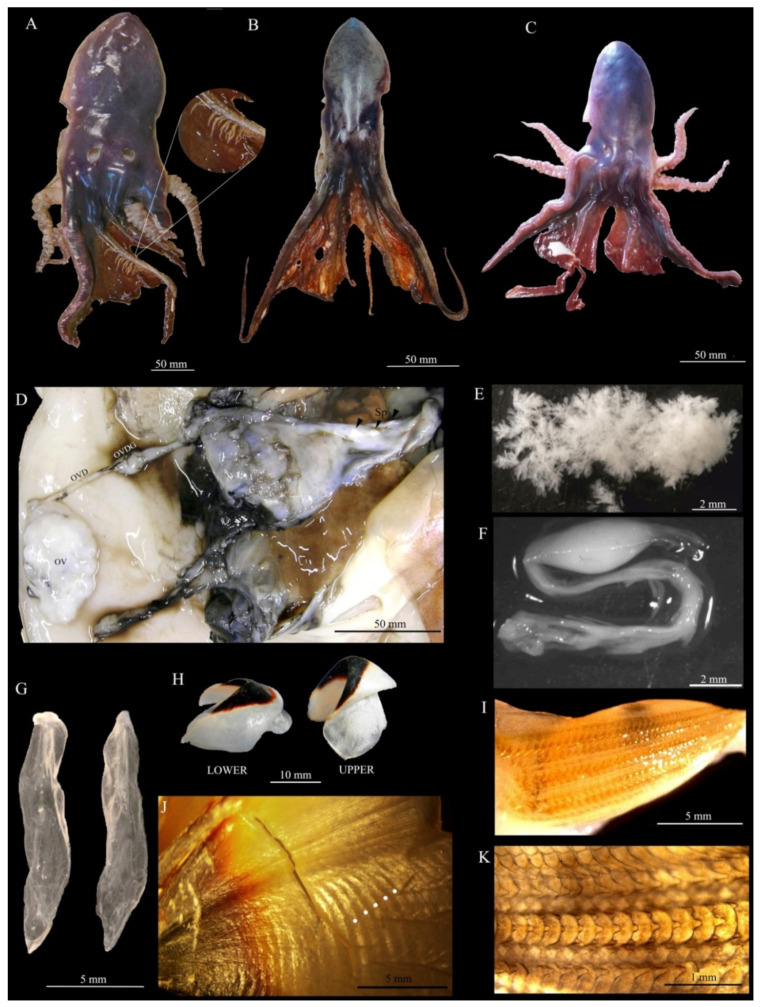
*Tremoctopus violaceus* from the Mediterranean Sea. Photographic records of the three females TV11 (**A**), TV16 (**B**), and TVBA (**C**) are shown. The magnified circle in (**A**) shows patches forming clusters arranged transversely in a piece of the web. Mantle cavity (**D**), ovary oocytes (**E**), and spermatangia (**F**) from TV11 are shown. Stylets from TV16 (**G**), upper and lower beak from TV11 (**H**), increments marked by points (**J**), and radula (**I**,**K**) from TV16 are shown. Abbreviations: OV, ovary; OVD, ovidutcts; OVDG, oviducal gland; Sp, spermatangium.

**Figure 2 animals-12-00080-f002:**
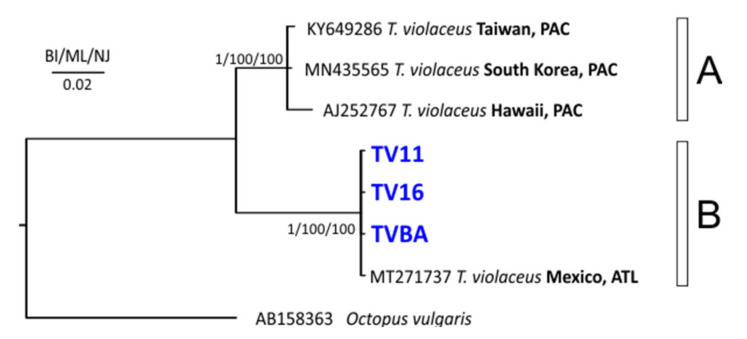
Phylogenetic tree obtained with the 16S sequences. Near the nodes are the support values for the Bayesian interference (BI), maximum likelihood (ML), and neighbor-joining (NJ) methods. PAC, Pacific Ocean; ATL, Atlantic Ocean. Appendix A contains all details and references of the sequences used. In blue are the *Tremoctopus* sequences from the Mediterranean Sea. Next to the tips are the GenBank accession numbers, under which the sequences are deposited in the repository, and their geographical origins. The letters near the tree (A and B), indicate the clades.

**Figure 3 animals-12-00080-f003:**
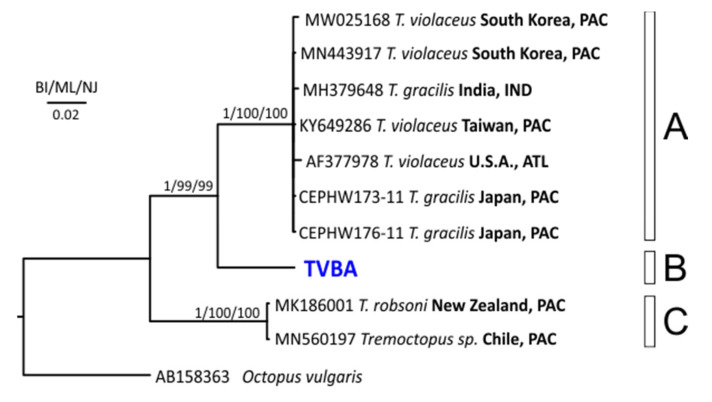
Phylogenetic tree obtained with the cytochrome c oxidase subunit I (COI) sequences. Near the nodes are the support values for the BI, ML, and NJ methods. PAC, Pacific Ocean; IND, Indian Ocean; ATL, Atlantic Ocean. Appendix A contains all details and references of the sequences used. In blue is the *Tremoctopus*’ sequence from the Mediterranean Sea. Next to the tips are the GenBank accession numbers, under which the sequences are deposited in the repository, and their geographical origins. The letters near the tree (A, B and C), indicate the clades.

**Table 1 animals-12-00080-t001:** Features of the three specimens of *Tremoctopus violaceus* caught in the Mediterranean Sea.

Sample code	TV11	TV16	TVBA
Locality	Sardinian waters	Sardinian waters	Adriatic Sea
Date	15 September 2011	24 November 2015	16 June 2017
Total weight	875.0	93.0	106.0
Sex	Female	Female	Female
Dorsal mantle length	165.0	81.5	76.6
Ventral mantle length	100.0	53.2	62.5
Mantle width (index %)	130.0 (78.8)	48.5 (59.5)	49.9 (65.1)
Head length (index %)	61.0 (37.0)	22.6 (27.7)	28.1 (36.7)
Head width (index %)	105.0 (63.6)	44.6 (54.7)	40.5 (52.9)
Dorsal pore size (left/right)	15.5 × 11.8/15.5 × 10.1	7.6 × 4.6/7.2 × 4.6	7.4 × 5.1/7.2 × 4.9
Ventral pore size (left/right)	8.8 × 5.1/8.8 × 4.9	6.5 × 4.3/6.1 × 4.3	4.5 × 3.8/4.1 × 3.6
Arm length I (left/right)	80 **/84 **	128 */127 *	92 */103 *
Arm length II (left/right)	-/140 **	231/241	159 */189
Arm length III (left/right)	86 **/88 **	99/98	93/90
Arm length IV (left/right)	70 **/90 **	137/137	114/113
Number gill lamellae	13	13	13
Maturity condition	Maturing (mated)	Immature	Developing
Ovary oocytes size (median)	0.06–0.70 (0.35)	0.14–0.46 (0.26)	0.16–0.41 (0.26)
Ovary potential fecundity	105,758	20,140	11,237
Stylets size	12.81 × 2.20	12.78 × 2.45	10.10 × 2.00
Crest length (upper/lower beaks)	16.9/11.03	9.37/5.28	9.13/5.71
Hood length (upper/lower beaks)	12/8.27	5.53/2.53	5.05/3.18
Wing length (upper/lower beaks)	8.22/10	4.38/3.8	4.31/4.0
LWA (upper beak)	13.2	6.16	7.17
JAD (upper/lower beaks)	9.52/7.3	4.33/4.45	5.16/4.72
BL (lower beak)	13.8	9.53	9.31
Stomach fullness index	0	1	1
Beak increments (mean ± SD; unit: day) Coefficient of variation (%) Average percent error	161 ± 0.84 0.1 0.40	92 ± 1.40 0.1 0.87	74 ± 1.00 0.1 1.08

The measurements results are shown in mm; the weight is expressed in g; * truncated; ** heavily damaged; -, not determined.

**Table 2 animals-12-00080-t002:** Genetic distances among the *Tremoctopus* groups identified in the trees, calculated in MEGA in the Kimura two-parameter model [36]. Intra-group distances are shown in the diagonal in italic, while inter-group distances are shown below the diagonal.

COI	Clade A	Clade B	Clade C
Clade A	*0.00198*		
Clade B	0.08867	na	
Clade C	0.14282	0.14661	*0.00000*
16S	Clade A	Clade B	
Clade A	*0.00576*		
Clade B	0.06393	*0.00000*	

## Data Availability

The data presented in this study are available from the corresponding author on request.

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
