# Peer review of "First Integrative Morphological and Genetic Characterization of Tremoctopus violaceussensu stricto in the Mediterranean Sea"

_animals, 2021, doi:10.3390/ani12010080_

Round 1
Reviewer 1 Report
The work is interesting. What I do not see is the importance of clarifying the species correct is which was never doubted before.
Suggestions:
Introduction
According to WoRMS the correct authorship for Tremoctopus gracilis has as follows:Tremoctopus gracilis (Souleyet, 1852)
Tremoctopus violaceus gracilis (Souleyet, 1852) has been accepted as Tremoctopus gracilis (Souleyet, 1852) an alien species in the Mediterranean Sea reported from Tunisia (Rifi & Ben-Souissi, 2014/ Ounifi-Ben Amor et al., 2016); Italy (Orsi Relini et al., 2004); Croatia (Kramer, 1937). An extensive discussion on its status is given in Bello et al (2020)
Rifi, M. & J. Ben Souissi. 2014. Première mention du poulpe palmée Tremoctopus gracilis (Eydoux and Souleyet, 1852) dans le Golfe de Tunis. In: Proceedings du 4ème congrès Franco-Maghrébin et 5èmes journées FrancoTunisiennes de Zoologie. Korba, Tunisia, 19
Ounifi-Ben Amor, K., Rifi, M., Ghanem, R., Draeif, I., Zaouali, J., Ben Souissi, J., 2016. Update of alien fauna and new records from Tunisian marine waters. Mediterr. Mar. Sci. 17 (1), 124e143. http://dx.doi.org/10.12681/mms.1371.
Orsi Relini L., Belluscio A. & Ardizzone G.D., 2004. Tracking the Indopacific pelagic octopus Tremoctopus gracilis in the Mediterranean. Rapports et procès verbaux de la Commission Internationale pour l’exploration scientifique de la Mer Méditerranée, 37: 415.
Bello, G., Andaloro, F., & Battaglia, P. (2020). Non-indigenous cephalopods in the Mediterranean Sea: a review. Acta Adriatica: International Journal of Marine Sciences, 61(2), 113-134.
For discussion
Tremoctopus gracilis is considered an established alien Species in the Mediterranean (Zenetos & Galanidi, 2020). Uncertainties of the correct identification of previously reported records cast doubts on a) its presence b) establishment success in the Mediterranean.
Zenetos, A., & Galanidi, M. (2020). Mediterranean non indigenous species at the start of the 2020s: recent changes. Marine Biodiversity Records, 13(1), 1-17.
References 1, 2 & 3 can be replaced with WoRMS Editorial Board, 2021
Author Response
We thank the reviewer 1 for giving us the opportunity to improve our work
A check of the English language has been done by a mother tongue
Comments and Suggestions for Authors
The work is interesting. What I do not see is the importance of clarifying the species correct is which was never doubted before.
Replay: We thank the referee for this comment. Considering the uncertainties on the correct morphological identification previously reported within this genus and the reported occurrence in the Mediterranean waters of the congeneric species T. gracilis, the authors aimed to use an integrative approach based on morphological and molecular analysis to correctly identify our investigated specimens.
Suggestions:
Introduction
According to WoRMS the correct authorship for Tremoctopus gracilis has as follows:Tremoctopus gracilis (Souleyet, 1852)
Replay: Thank you, in the revised version we have insert the correct autorship according to WoRMS (line 48).
Tremoctopus violaceus gracilis (Souleyet, 1852) has been accepted as Tremoctopus gracilis (Souleyet, 1852) an alien species in the Mediterranean Sea reported from Tunisia (Rifi & Ben-Souissi, 2014/ Ounifi-Ben Amor et al., 2016); Italy (Orsi Relini et al., 2004); Croatia (Kramer, 1937). An extensive discussion on its status is given in Bello et al (2020)
Rifi, M. & J. Ben Souissi. 2014. Première mention du poulpe palmée Tremoctopus gracilis (Eydoux and Souleyet, 1852) dans le Golfe de Tunis. In: Proceedings du 4ème congrès Franco-Maghrébin et 5èmes journées FrancoTunisiennes de Zoologie. Korba, Tunisia, 19
Ounifi-Ben Amor, K., Rifi, M., Ghanem, R., Draeif, I., Zaouali, J., Ben Souissi, J., 2016. Update of alien fauna and new records from Tunisian marine waters. Mediterr. Mar. Sci. 17 (1), 124e143. http://dx.doi.org/10.12681/mms.1371.
Orsi Relini L., Belluscio A. & Ardizzone G.D., 2004. Tracking the Indopacific pelagic octopus Tremoctopus gracilis in the Mediterranean. Rapports et procès verbaux de la Commission Internationale pour l’exploration scientifique de la Mer Méditerranée, 37: 415.
Bello, G., Andaloro, F., & Battaglia, P. (2020). Non-indigenous cephalopods in the Mediterranean Sea: a review. Acta Adriatica: International Journal of Marine Sciences, 61(2), 113-134.
Replay: On the base of the references suggested the authors have improved the introduction taking into account the status of the genus in Mediterranean, including the species Tremoctopus gracilis accepted as an alien species (see new lines 57-85).
All the references suggested are now cited in the text of the manuscript and reported in the references list (see number 17-18-19-20)….
For discussion
Tremoctopus gracilis is considered an established alien Species in the Mediterranean (Zenetos & Galanidi, 2020). Uncertainties of the correct identification of previously reported records cast doubts on a) its presence b) establishment success in the Mediterranean.
Replay: In the amended version of the manuscript, we clearly stated that T. gracilis is considered an established alien species in the Mediterranean and we also added the suggested references in the text. Certainty, an integrative approach also based on molecular tools, is needed in order to avoid misidentification and provide reliable data regarding its presence, distribution and establishment success in the basin. Regarding this point we have added a comment in the introduction and in the conclusion (see new lines 405-408)
Zenetos, A., & Galanidi, M. (2020). Mediterranean non indigenous species at the start of the 2020s: recent changes. Marine Biodiversity Records, 13(1), 1-17.
We have inserted in the text (line 75) and in the references list (N°20
References 1, 2 & 3 can be replaced with WoRMS Editorial Board, 2021
Replay: Sorry, but we don’t understand why we have to replace these references considering that are cited also in different parts of the manuscript.

Reviewer 2 Report
Major comments:
I think this paper should be published, but my reason does not match what the authors indicate in the title and text as their reason, documenting the occurrence of Tremoctopus violaceus in the Mediterranean. Thus the emphasis and importance of the manuscript should be rewritten.
Although Chiaie (1830 In 1823-1831) did not formally designate a type locality for T. violaceus, the publication was on invertebrates of the Kingdom of Naples. It is safe to consider the type locality to be Mediterranean. It is therefore not surprising that this species was found in the Med.
It has become very obvious in recent years that many species, such as this one, considered to be global in distribution may actually be species complexes. If not cryptic complexes of species, then some widely distributed species have substantial population structure reflected in their genetic sequences. It is therefore important, when considering a gene sequence to be typical of a species, to use a sequence from a specimen caught as close as possible to the type locality of the species.
However, of 12 public entries for T. violaceus in GenBank, I could find no indication that any were from the Med, or even from the eastern North Atlantic. I therefore think that this documentation of barcode sequences important for species determination from the presumed type locality is important and should be published. An example of the utility of this is the authors' comparison of the sequences from the Med with those of specimens published from the Gulf of Mexico indicating that the latter are indeed conspecific with the "true" T. violaceus.
The authors describe morphological characters for females of the species from the type locality. They also properly question the importance of color patterns for defining species of Tremoctopus.
Minor comments:
l. 45 -- Tremoctopus gelatus is presumably mesopelagic, not epipelagic.
Although not bad, the English needs work.
Author Response
We thank the reviewer 2 for giving us the opportunity to improve our work
Comments and Suggestions for Authors
Major comments:
I think this paper should be published, but my reason does not match what the authors indicate in the title and text as their reason, documenting the occurrence of Tremoctopus violaceus in the Mediterranean. Thus the emphasis and importance of the manuscript should be rewritten.
Replay: The authors has modified the Title and the text, implementing the Introduction (see new lines 74-94) and the Conclusions (405-408), also taking into account the considerations and comments of the reviewer1, to emphasise the importance of the combined approach of molecular and morphological tools, for the characterization of the species T. violaceus sensu stricto, in the Mediterranean Sea, that is considered the type locality of the species.
Although Chiaie (1830 In 1823-1831) did not formally designate a type locality for T. violaceus, the publication was on invertebrates of the Kingdom of Naples. It is safe to consider the type locality to be Mediterranean. It is therefore not surprising that this species was found in the Med.
It has become very obvious in recent years that many species, such as this one, considered to be global in distribution may actually be species complexes. If not cryptic complexes of species, then some widely distributed species have substantial population structure reflected in their genetic sequences. It is therefore important, when considering a gene sequence to be typical of a species, to use a sequence from a specimen caught as close as possible to the type locality of the species.
However, of 12 public entries for T. violaceus in GenBank, I could find no indication that any were from the Med, or even from the eastern North Atlantic. I therefore think that this documentation of barcode sequences important for species determination from the presumed type locality is important and should be published. An example of the utility of this is the authors' comparison of the sequences from the Med with those of specimens published from the Gulf of Mexico indicating that the latter are indeed conspecific with the "true" T. violaceus.
Replay: Thank you to the review. Actually, no genetic analyses have been conducted to date in the Mediterranean Sea about the genus Tremoctopus, and no sequences are available in the repositories from this area to compare with our data.
The authors describe morphological characters for females of the species from the type locality. They also properly question the importance of color patterns for defining species of Tremoctopus.
Minor comments:
- 45 -- Tremoctopus gelatus is presumably mesopelagic, not epipelagic.
Replay:– Yes thanks. We have modified the phrase (see new line 47)
Although not bad, the English needs work.
Replay: A check of the English language has been done by a mother tongue
Reviewer 3 Report
The paper sounds interesting but the experimental design it is not clear and the number of samples examined are not sufficient to determine a new species.
I suggest to the Authors to improve the experimental design especially the molecular section and increase the number of samples
Author Response
Comments and Suggestions for Authors
The paper sounds interesting but the experimental design it is not clear and the number of samples examined are not sufficient to determine a new species.
I suggest to the Authors to improve the experimental design especially the molecular section and increase the number of samples
Replay: The authors would like to have more samples to analyse but unfortunately as already emphasized in different parts of the manuscript the species of the genus Tremoctopus are rare. Indeed, our samples were found sporadically (2011, 2016, 2018) and given to the research institute by citizens.
We would like to underline that the aim of our work is not to determinate a new species but to use an integrative approach based on combined morphological and molecular analysis that give certainties on the species belonging of our samples, considering that in Mediterranean Sea occur two species of genus Tremoctopus (T.violaceus and T. gracilis). Consequently, all the biological information given in this work (reproduction, age, feeding) can be certanly associated to Tremoctopus violaceus sensu stricto.
A check of the English language has been done by a mother tongue
Round 2
Reviewer 2 Report
I did not find this revision to be substantially improved over the previous version. Most of my previous comments still apply.
Author Response
Open Review2
Comments and Suggestions for Authors
Major comments:
I think this paper should be published, but my reason does not match what the authors indicate in the title and text as their reason, documenting the occurrence of Tremoctopus violaceus in the Mediterranean. Thus the emphasis and importance of the manuscript should be rewritten.
Reply: the main objective of the study as well as the title, introduction, results, discussion, and conclusions have been further changed.
Although Chiaie (1830 In 1823-1831) did not formally designate a type locality for T. violaceus, the publication was on invertebrates of the Kingdom of Naples. It is safe to consider the type locality to be Mediterranean. It is therefore not surprising that this species was found in the Med.
Reply: We were not surprised by the finding of T. violaceus in the Mediterranean, but, considering the many recent records of a similar alien species (T. gracilis), we would like to be sure of the identification of our samples, so we have analyzed them not only morphologically but also molecularly.
It has become very obvious in recent years that many species, such as this one, considered to be global in distribution may actually be species complexes. If not cryptic complexes of species, then some widely distributed species have substantial population structure reflected in their genetic sequences. It is therefore important, when considering a gene sequence to be typical of a species, to use a sequence from a specimen caught as close as possible to the type locality of the species.
Reply: we fully agree with the comment of the reviewer. It is now explicitly written in the text that our sequences are from the type locality of T. violaceus.
However, of 12 public entries for T. violaceus in GenBank, I could find no indication that any were from the Med, or even from the eastern North Atlantic. I therefore think that this documentation of barcode sequences important for species determination from the presumed type locality is important and should be published. An example of the utility of this is the authors' comparison of the sequences from the Med with those of specimens published from the Gulf of Mexico indicating that the latter are indeed conspecific with the "true" T. violaceus.
Reply: we fully agree with the comment of the reviewer. It is now emphasized that new barcoding sequences are available for the species.
The authors describe morphological characters for females of the species from the type locality. They also properly question the importance of color patterns for defining species of Tremoctopus.
Reply: we fully agree with the comment of the reviewer. It is now emphasized in the text the need of be cautious when using the web colour pattern to discriminate between T. violaceus and T. gracilis
Although not bad, the English needs work.
Reply: the manuscript has been further checked and corrected for style and grammar by a mother tongue English teacher.